# Sub-resolution contrast in neutral helium microscopy through facet scattering for quantitative imaging of nanoscale topographies on macroscopic surfaces

Sabrina D. Eder [1,4], Adam Fahy [2,4], Matthew G. Barr[2,4], J. R. Manson [3], Bodil Holst [1] & Paul C. Dastoor [2]✉

Nanoscale thin film coatings and surface treatments are ubiquitous across industry, science, and engineering; imbuing specific functional or mechanical properties (such as corrosion resistance, lubricity, catalytic activity and electronic behaviour). Non-destructive nanoscale imaging of thin film coatings across large (ca. centimetre) lateral length scales, crucial to a wide range of modern industry, remains a significant technical challenge. By harnessing the unique nature of the helium atom–surface interaction, neutral helium microscopy images these surfaces without altering the sample under investigation. Since the helium atom scatters exclusively from the outermost electronic corrugation of the sample, the technique is completely surface sensitive. Furthermore, with a cross-section that is orders of magnitude larger than that of electrons, neutrons and photons, the probe particle routinely interacts with features down to the scale of surface defects and small adsorbates (including hydrogen). Here, we highlight the capacity of neutral helium microscopy for sub-resolution contrast using an advanced facet scattering model based on nanoscale features. By replicating the observed scattered helium intensities, we demonstrate that sub-resolution contrast arises from the unique surface scattering of the incident probe. Consequently, it is now possible to extract quantitative information from the helium atom image, including localised ångström-scale variations in topography.

Neutral helium microscopy exploits the inherent properties of its probe particle[1] (inert, low polarisability, no net spin, and a de Broglie wavelength of the order of typical crystallographic dimensions at milli-electron volt energies) to produce an imaging technique ideal for materials typically degraded under the energetic probes of other microscopies[2–4]. As the probe cannot penetrate the bulk at all, the micrograph generated is exclusively of the surface under investigation[5]. The scanning helium microscope (SHeM) collimates a free-jet expansion of neutral helium via a simple pinhole aperture, defining the lateral resolution of the instrument[6]. Currently, resolution is primarily limited by detector efficiency, with state-of-the-art detectors enabling resolutions of the order of 40 nm[7]. By rastering the sample underneath the beam and collecting the backscattered helium signal, an image may be generated.

[1]Department of Physics and Technology, University of Bergen, Bergen, Norway. [2]Centre for Organic Electronics, University of Newcastle, Callaghan, NSW 2308, Australia. [3]Department of Physics and Astronomy, Clemson University, Clemson, SC 29634, USA. [4]These authors contributed equally: Sabrina D. Eder, Adam Fahy, Matthew G. Barr. ✉e-mail: paul.dastoor@newcastle.edu.au

The nature of the atom–surface interaction[8] dictates the contrast mechanisms available in the generated micrographs. Neglecting resonant state trapping (unlikely for neutral helium atoms[9]), the various interactions can be categorised as either elastic or inelastic processes. The dominant contrast mechanism is topographic in nature, which is itself a subset of the possible elastic scattering trajectories. The localised tilt of the sample surface with respect to the detector determines the collected signal, thus allowing the surface morphology to generate image contrast through 'masking' and 'shadowing' (beam and detector occlusion, respectively)[6,10]. Previous work has focused on surfaces with feature sizes greater than the lateral resolution of the instrument (supra-resolution), whereby the image formation mechanism results in the direct observation of these features[6]. However, the same scattering phenomena must also occur for feature sizes smaller than the lateral resolution of the instrument (sub-resolution). In addition, the reflected intensity can also be influenced by other elastic processes (such as multiple scattering events[6,10,11] and, in the case of crystalline materials, diffraction[12]) or inelastic processes whereby the composition and local atomic character of the sample surface can also give rise to differences in the helium reflectivity[13]; resulting in distinct changes in Michelson contrast. Therefore, in principle, the SHeM is capable of generating contrast from sub-resolution topographic features in circumstances where this contrast channel dominates. The sensitivity of neutral helium to low concentrations of surface defects and adsorbates is well-documented in helium atom scattering (HAS) literature[14]. However, to date there have been no systematic investigations into the sensitivity of SHeM to such sub-resolution features.

## Results and discussion
### Imaging of ultrathin films
As a first step, model thin film samples were prepared to establish whether the SHeM can generate sufficient contrast to image sub-resolution structures at ultralow concentrations. Utilising a thermal evaporator, patterned ultrathin gold films of different thicknesses were deposited onto n-doped silicon surfaces (comprising a $SiO_2$ layer) and imaged in the SHeM (see Methods for complete fabrication and imaging details). It should be noted that there was no subsequent sample surface modification (cleaning or special coating) post evaporation. Figure 1(a–c) shows SHeM micrographs of $(9 \pm 1)$, $(80 \pm 8)$ and $(778 \pm 78)$ Å thick patterned gold films on silicon, with a clear contrast difference between the underlying substrate and the patterned gold layer for all three thicknesses. Moreover, the degree of contrast between the silicon and the gold layers varies across the three samples, demonstrating that the SHeM can differentiate between the coverages even though all three film thicknesses are orders of magnitude below the lateral resolution (~1 μm) and minimum observable step height (~70 μm) of the instrument[6]. The averaged linescans in Fig. 1(d–f) highlight that the measured contrast does not change monotonically with thickness, but rather is greatest for the 80 Å film.

### Facet-scattering model
In order to establish whether the non-monotonic contrast variations with gold thickness (observed in the SHeM micrographs shown in Fig. 1) arise from sub-resolution topographic features, a bespoke helium atom–facet-scattering MATLAB simulation was developed. The facet-scattering model (detailed in Methods) utilises height maps, such as those generated by an atomic force microscope (AFM), to calculate the local tilt at each pixel via nearest neighbour interpolation. Shadowing, masking, and the local facet angle are used to determine whether a scattered helium atom enters the detector. Calculating the detector acceptance for each facet thus yields a relative detected helium intensity per unit area. Provided that the input height maps span an area of the order of the lateral resolution of the SHeM and are a good approximation of the surface under investigation, the output

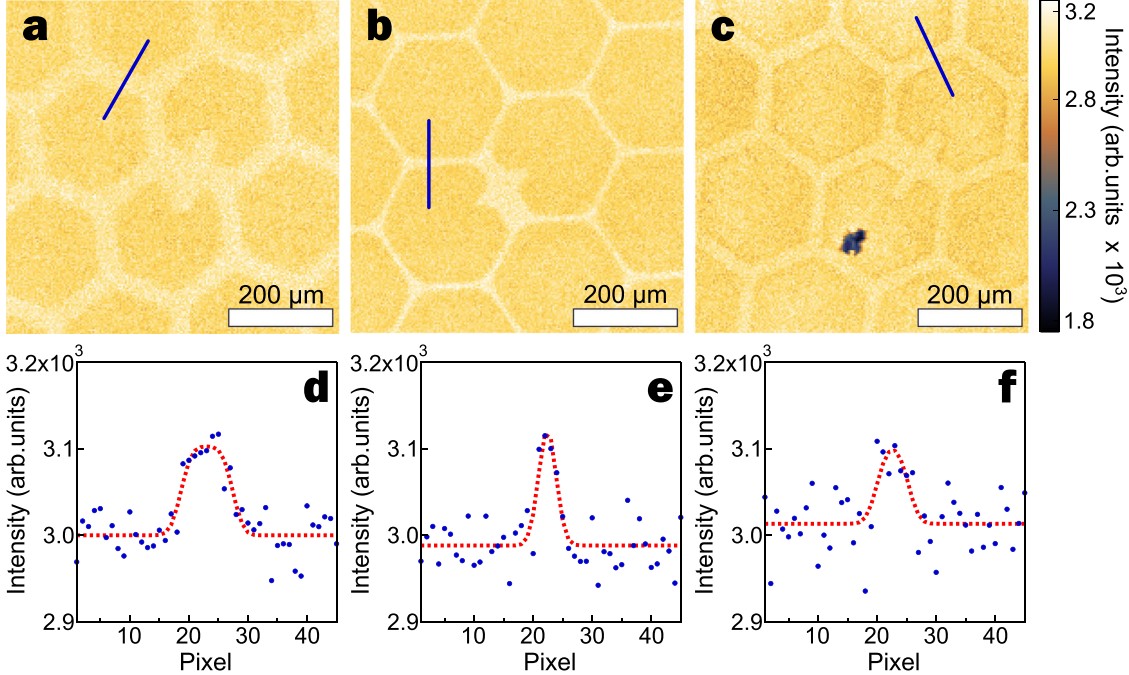

**Fig. 1 | Patterned gold films on silicon substrate.** SHeM micrographs of thermally evaporated gold films of thickness **a** $9 \pm 1$; **b** $80 \pm 8$; and **c** $778 \pm 78$ Å. The substrate was masked by a TEM grid, resulting in unaffected silicon (bright pattern) and the gold film (darker regions). The dark spot in **c** is a dust particle, exhibiting supra-resolution topographic contrast. A different TEM grid batch was used for micrograph **b**. All micrographs use the same intensity range (see colorbar) to allow direct comparison of the available contrast and are normalised to the intensity of the regions of gold film in micrograph **a**. All micrographs have been collected with a 4 μm step between pixels, and scale bars are 200 μm in length. **d**–**f** show averaged linescans across the masked regions for micrographs **a**–**c**, respectively, with example lines shown in the micrographs. Dashed red line for each linescan has been plotted to help guide the eye.

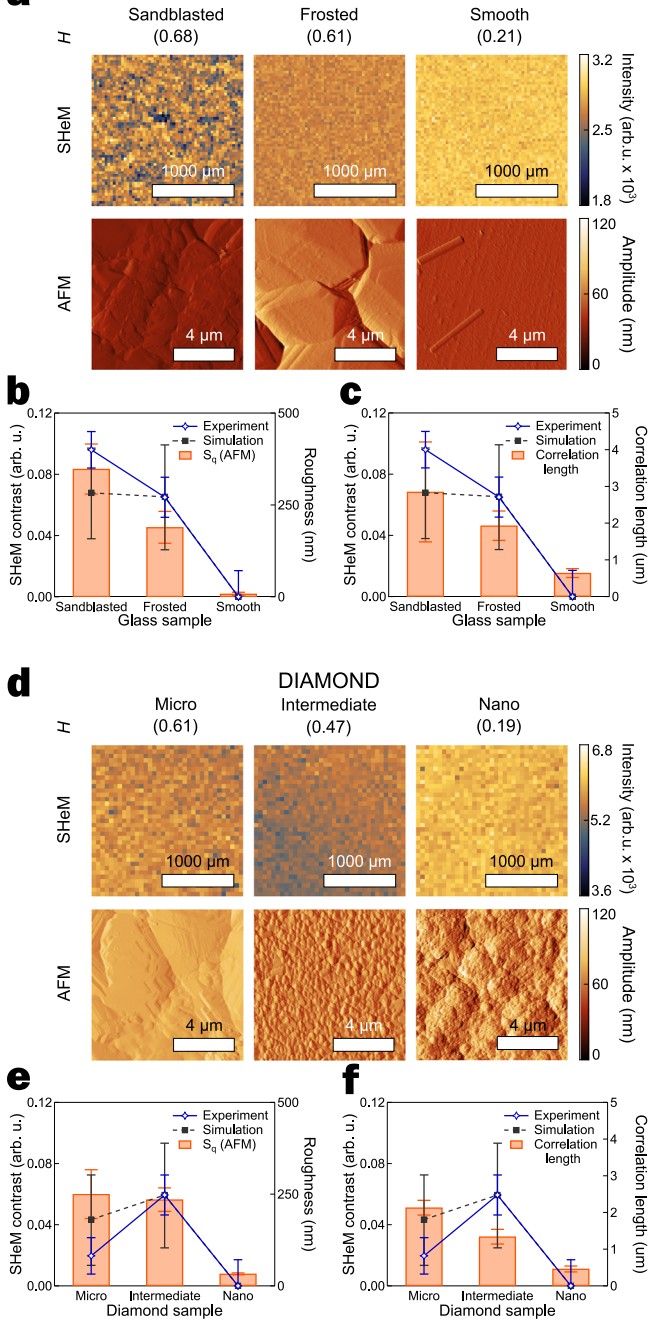

**Fig. 2 | Comparison of facet-scattering model and experiment.** AFM (amplitude) and SHeM micrographs of: **a** glass and **d** diamond surfaces for a range of surface finishes (summarised by the Hurst parameter $H$). AFM scale bars are 4 μm; SHeM scale bars are 1000 μm. **b** and **e** Experimental SHeM Michelson contrast values (open stars) for the glass and diamond sample sets (respectively) as derived from the micrographs shown in **a** and **d**, along with simulation results (solid squares) for the same surfaces. Diamond surface contrast referenced to a silicon wafer; glass surface contrast taken with respect to the smooth (unaltered) glass sample. Errors in the experimental SHeM contrast are estimated using the standard deviation of the micrograph intensities and standard error propagation. Standard deviations for the simulated micrograph intensities were estimated via rotating the input map (see Methods), with the resultant errors for contrast following from standard error propagation. Areal RMS roughness ($S_q$) was derived from the AFM micrographs of each surface (orange bar chart), with associated errors found by calculating the 2D RMS roughness for 5 different sub-regions of the AFM map and determining the standard deviation. **c** and **f** Comparison of the experimental and simulation contrast results with the correlation length as derived from the AFM micrographs of each surface (orange bar chart). The correlation length error bars represent the 95% confidence interval for the calculation (see Supplementary Discussion 2 for details).

intensities represent the expected contrast arising from elastically scattered helium in the SHeM micrographs.

The model was first validated using single component surfaces of glass and diamond; in principle removing the possibility for inelastic scattering contrast arising from the presence of different materials[8]. Various surface treatments (on glass samples) and growth conditions (for diamond samples) were used to generate model surface topographies spanning a broad range of feature sizes below the lateral resolution of the instrument; i.e., the nano to micro length scales (see Methods section for full details). The Hurst parameter $H$ (see Supplementary Discussion 2) describes the nature of the 'roughness' of a data series; the smaller the value of $H$ the less long-range order and vice versa. When $H$ is exactly 0.5, the data can be described by geometric Brownian motion; $H < 0.5$ corresponds to little surface waviness, ie. a jagged surface profile; and $H > 0.5$ indicates longer range order such as faceting[15]. AFM height maps (Fig. 2a, d and Supplementary Discussion 2) confirmed that the surface features spanned a broad range of Hurst parameters consistent with the desired surface topographies. Finally, beam energy studies performed on the SHeM, as well as energy-resolved HAS time-of-flight (TOF) measurements, verified elastic scattering is the dominant scattering channel for both sets of materials (see Methods and Supplementary Discussion 1).

Figure 2a, d present the SHeM micrographs of the glass and diamond surfaces, respectively. The SHeM micrographs are predominantly featureless but show significant intensity differences between the single component surfaces, consistent with the observed contrast arising from variations in surface topography below the lateral resolution of the instrument. The exception is the sandblasted glass sample, which exhibits distinct regions of experimental contrast associated with facet-like features ($H > 0.5$) above the lateral resolution of the SHeM.

As the AFM maps are of the order the helium spot size on the sample, a possible explanation for the observed differences in SHeM images is that the sub-resolution contrast arises simply from localised variations in 2D RMS roughness ($S_q$). Figure 2b, e compare the simulated and experimental SHeM contrast to $S_q$ as a function of length scale for both glass and diamond systems, respectively. The relative SHeM contrast was confirmed using complementary HAS measurements (see Methods section and Supplementary Discussion 1). While the RMS roughness values calculated from the AFM data correlate with the SHeM image contrast for the glass dataset, they are wholly insufficient in explaining the diamond dataset. Higher order moments (such as skewness and kurtosis) were also found to be incapable of explaining the observed trends in SHeM contrast. As such, the dependence of the image formation process upon surface topography cannot be explained by simple statistical metrics[16]. For example, the $S_q$ values shown in Fig. 2e indicate that the micro-diamond surface should yield the most contrast; yet the intermediate-diamond surface appeared darkest in SHeM micrographs. More sophisticated analysis methodologies such as the Hurst parameter and the correlation length (Fig. 2c, f) were also unable to explain the SHeM contrast for the diamond system (see Supplementary Discussion 2 for the specific calculation).

By comparison, there is good agreement between the simulated and experimental SHeM contrast for both materials systems, demonstrating that image contrast is a direct result of the interplay between the helium beam and the specific sample topography. As it is both the distribution of surface features and the scattering geometry working in concert that yield the contrast apparent in SHeM micrographs, only the facet-scattering model can replicate the intricacies of the image formation process. Moreover, the observed trends in the SHeM contrast data for the single component surfaces are more complex than simple roughness metrics and statistical moments indicate.

The robustness of the model was tested in two ways. First, surfaces exhibiting supra-resolution contrast are expected to break the

fundamental assumption underpinning the modelling, namely that the AFM maps are representative of the entire region being imaged. The model tests for such conditions by rotating the input height map and simulating the expected relative intensity for each orientation. Any significant disparity between these simulations indicates the presence of supra-resolution features (see Methods). In addition, any large variations in modelled intensity between different input AFM datasets for the same sample also indicate that supra-resolution features are the dominant contributor to contrast. These conditions were only observed for the sandblasted glass sample (intensity differences >700% between regions) and consequently this sample exhibited the poorest agreement between simulation and experimental SHeM contrast. Second, when a 2D averaging filter or matrix downsampling was applied to the AFM maps prior to running the simulations (in an attempt to reduce computational runtime), the predictive power of the model was destroyed in every case. This image processing is analogous to collecting AFM micrographs with a blunt tip. We therefore conclude that whilst neutral helium (and thus the SHeM) sees surface topography at least to the scale that is probed by AFM, it is the very fundamentals of image formation from elastically scattered helium—in particular shadowing, masking and facet angle—that are the dominant factors in determining SHeM contrast.

## Characterisation of gold films

Having validated the facet-scattering model in simulating SHeM contrast from sub-resolution topographic features in single component model surfaces, it could now be applied to SHeM analysis of the thin gold film systems such as those shown in Fig. 1. Polycrystalline gold deposited onto SiO$_2$ via thermal evaporation is a complicated thin film growth system[17]. The growth exhibits distinct primary and secondary grain formation stages[18,19]. Initially, growth occurs through the formation of primary grains with a size of the order of the film thickness, with the irregular open channels between grains gradually filling in as deposition continues. Large secondary grains then form spontaneously once the first layer is completed. To determine the capability of the SHeM to probe sub-resolution topography, a series of polycrystalline gold films of mean thickness 1 Å (that is, less than a monolayer) up to approximately 800 Å were deposited onto SiO$_2$ substrates via thermal evaporation (see Methods). The gold films were shadow masked, leaving regions of uncovered SiO$_2$ substrate for contrast measurements. The gold films were comprehensively characterised using SHeM, AFM and scanning electron microscope (SEM) imaging with the corresponding micrographs—as well as the experimentally determined SHeM contrast values—for each thin film thickness shown in Fig. 3. As evidenced by the normalised SHeM micrographs, the

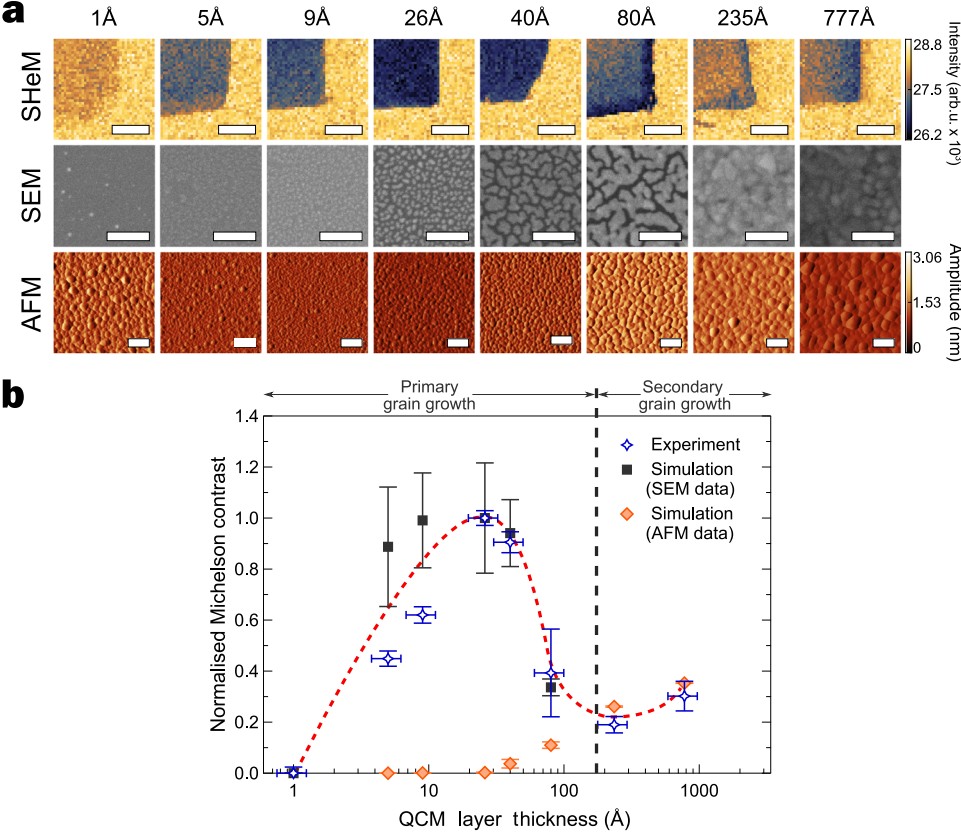

**Fig. 3 | Experimental and modelled SHeM contrast for thermally evaporated gold layers on silicon. a** AFM (amplitude), SEM and SHeM micrographs of the manufactured gold-on-silicon samples, with the quartz crystal microbalance (QCM) thickness of the evaporated gold indicated along the top. AFM and SEM images show specifically the gold region of the sample in order to illustrate the development of surface features according to the growth mode of the material. Such features form the basis for the masking and shadowing responsible for the observed SHeM contrast. SHeM micrographs show a gold-silicon interface, with the gold being the darker material. Scale bars for the AFM and SEM micrographs are 100 nm in length; SHeM scale bars 400 μm in length. All AFM images have been restricted to the same amplitude range (derived from the maximum amplitude found across the total dataset, namely 0–3.06 nm) for direct comparison. SHeM micrographs have first been normalised to the silicon substrate, and restricted to the same intensity range (26.2–28.8 kHz) for direct comparison of the changing contrast. **b** Plot of the normalised (scaled to sit between 0 and 1) experimental Michelson contrast between gold and silicon derived from the SHeM micrographs of the sample series as a function of QCM layer thickness. The predicted Michelson contrast between the gold and silicon from the facet-scattering model is also shown, based on height maps derived from the SEM and AFM images, as well as a line to guide the eye (red dotted line). Errors in the experimental SHeM contrast are estimated using the standard deviation of the micrograph intensities and standard error propagation. Standard deviations for the simulated micrograph intensities were estimated via rotating the input map (see Methods), with the resultant errors for contrast following from standard error propagation.

SHeM contrast varies strongly as a function of deposition thickness, with a peak contrast value at a gold thickness of ca. 26 Å. Furthermore, there is a distinct change in contrast behaviour that occurs at the well-established onset of secondary grain growth (thickness ca. 175 Å[18]).

In the primary grain growth regime, the SEM micrographs clearly show the expected morphology of the gold film whereas the AFM images do not. In particular, the deep cracks known to form during primary grain growth were not observed in the AFM images, likely due to the extreme aspect ratio of these irregular[20,21] channels as well as gold adatoms adhering to the AFM tip[22,23]. Consequently, facet-scattering model simulations based on the AFM height maps were incapable of describing the experimental SHeM contrast in the primary grain growth regime (Fig. 3). However, given that the growth in the primary grain growth regime occurs via single height islands, it is possible to convert the SEM images into height maps for the simulation. By comparison, in the secondary grain growth regime the AFM shows clear topographic structure since the deep channels fill in and broader secondary grains form; features well-suited to AFM height mapping. However, the unconstrained height of the secondary grains and the loss of Z-number contrast (as the cracks fill in and the system becomes more homogeneous) means that it is no longer possible to obtain reliable height maps from the SEM images. Thus, accurately reproducing the surface topography for the facet-scattering simulation model requires using the complementary quantitative height information available from both the SEM and AFM images for the primary and secondary grain growth regimes, respectively.

Figure 3 compares the experimental and simulated SHeM contrast in the primary and secondary regimes, demonstrating that the facet-scattering model replicates the trend in the experimental data. Thus, it is the specifics of the helium-surface interaction—in particular the elastic, single scattering that occurs from the outermost electronic corrugation potential—that enables the SHeM to image sub-monolayer coverages over lateral distances of millimetres or greater. In other words, whilst the SHeM has been demonstrated to exhibit image formation behaviour analogous to that of an SEM, the highly surface-sensitive nature of the interaction and its lack of penetration into the bulk is more reminiscent of AFM.

The unique understanding of surface topography offered by neutral helium microscopy has widespread application to nanoscale thin film coatings and surface treatments (essential to corrosion resistance[24], lubricity[25], catalytic activity[26,27] and electronic behaviour[28]). The non-destructive imaging of thin film coatings across large lateral length scales is a major technical challenge for modern industry[29,30]. In particular, nanoscale defect control is critically important to the lithographic processes that have driven the 50-year history of semiconductor scaling, requiring detection of nano to micro scale defects across large wafer areas[31].

## Reconstructed height maps

The extreme sensitivity of SHeM contrast to nanoscale topographic features, in combination with its non-damaging nature, enables sub-resolution defect detection across large areas. In particular, for a well-characterised material system, inversion of the derived image contrast function (Fig. 3) can be used to examine layer quality across the breadth of the sample. Although the relation between film thickness and Michelson contrast is non-unique, consistent with most reconstruction-based techniques used to evaluate nano-coatings and thin films (e.g., ellipsometry[32,33], interferometry[34] and computer tomography[35,36]) working within appropriate constraints (for example, a film thickness $\lesssim 20$ Å for the Au on Si system) allows a unique solution to be provided.

A basic confirmation of such a reconstruction was performed using standard AFM calibration gratings (Supplementary Discussion 3), before moving to the more industrially relevant example of thermally evaporated polycrystalline gold contacts on a silicon substrate (Fig. 4). Figure 4a shows the SHeM intensity maps for a gold thickness calibration bar together with two gold contacts of mean deposition thickness (as derived from the quartz crystal microbalance)

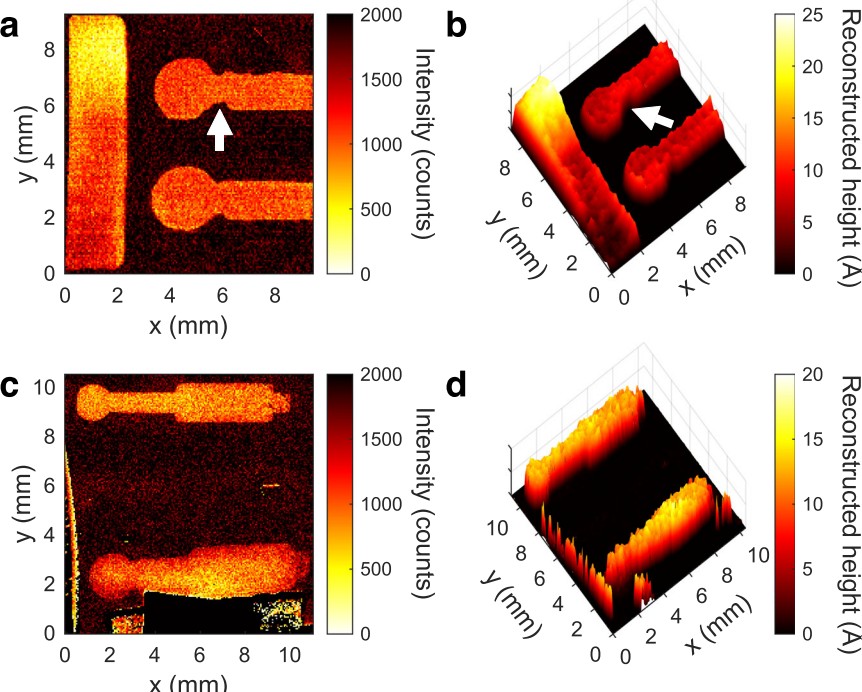

**Fig. 4 | SHeM micrographs and associated reconstructed height maps for gold contacts evaporated onto silicon wafer substrates. a** SHeM micrograph of gold thickness calibration bar with laterally defective (upper) and ideal (lower) gold contacts. **b** Corresponding reconstructed height map from **a**. **c** SHeM micrograph of ideal (upper) and vertically defective (lower) gold contacts. **d** Corresponding reconstructed height map from **c**. The white arrows correspond to locations referred to in the main text.

of ca. 12 Å. The shadow mask for the upper contact was deliberately designed to introduce lateral shape defects (white arrow) into the evaporated structure. Figure 4b shows the height map reconstructed from the SHeM intensities revealing that both contacts have excellent height uniformity (10 ± 2 Å) and that the lateral defects are clearly visible over their mm length scale. Figure 4c shows the SHeM intensity maps for two gold contacts, where the shadow mask for the lower contact was deliberately held away from the surface to introduce vertical shape defects into the evaporated structure. Figure 4d shows the reconstructed height maps revealing that the thickness of the defective contact varies relative to the 'ideal' upper contact, particularly at the edges of the film due to feathering from the offset shadow mask. Moreover, comparing the upper and lower contacts reveals readily observable deviations in mean height of ca. ±2–3 Å across lateral distances of millimetres, corresponding to aspect ratio sensitivities in excess of $10^8$. By comparison, a corresponding AFM image would require stitching of 250 000 individual (20 μm × 20 μm) images and (assuming 10 s per image) stable imaging for 29 days (694 h). Furthermore, as discussed earlier, AFM imaging in this gold coverage regime (10 ± 2 Å) is unreliable due to gold adatoms adhering to the AFM tip.

In summary, we have shown that the absolute surface sensitivity of neutral helium enables sub-resolution contrast in the SHeM to discern topographic features far below (and up to) the current lateral resolution of the instrument. The contrast observed in SHeM micrographs is driven by the processes of image formation: shadowing, masking, and facet angle. Incorporating these specific effects, a bespoke facet-scattering model was shown to describe the trends in the reflected intensities for multiple material systems, with feature sizes spanning orders of magnitude. Consequently, we showed that SHeM micrographs of sub-monolayer films can exhibit strongly varying contrast, arising from the specific nature of the sub-resolution scattering, thereby enabling ångström-scale height characterisation of thin films across centimetre lateral length scales. Its unique ability to observe the effects of sub-nanoscale features upon scattered helium intensity enables the SHeM to evaluate nano-coatings and thin films across large areas. Looking ahead, once a material system is well-characterised with SHeM, such analysis could readily become a routine part of quality control; a valuable tool for improving production yields and throughput.

## Methods

### Sample preparation: diamonds
Micro- and intermediate-diamond films were grown onto <100> Si substrates in a home-built 3 kW 2.45 GHz Microwave Plasma Chemical Vapour Deposition (MPCVD) reactor. Deposition for both samples was conducted in hydrogen-rich plasma for 4 h at a pressure of 150 Torr, keeping the surface temperature at 840 °C. The micro-crystalline diamond film was grown by introducing 7% $CH_4$ and 7% Ar into the chamber (86% $H_2$) while intermediate-diamond film morphology was achieved by interchanging Ar with 0.2% of $N_2$ (7% $CH_4$, 92.8% $H_2$). To produce the nanodiamond films, a Si wafer (500 μm thick, single side polished) was seeded via ultrasonication in a nanoparticle solution comprised of positively charged nanodiamond particles (PlasmaChem) and DI water (0.005 wt%). After seeding, the synthesis was carried out using a microwave plasma-assisted chemical vapour deposition (CVD) system (Seki Diamond Microwave System AX6500X). During the main step of deposition, a gas mixture of 97% $H_2$, 3% $CH_4$ was used at a pressure of 40 Torr and a power of 3000 W. A deposition time of 90 min was selected to achieve the desired diamond growth coverage and quality.

### Sample preparation: glass
A microscope glass slide (Livingston 7105-PPA) with frosted end was additionally sandblasted in-house (Econoline PB600SS sandblaster) on the opposite end to the frosted area. After sandblasting, the glass slide was cut with a diamond cutter to provide a sample of ca. 15 × 7 mm size piece including all three surface finishes (sandblasted, smooth, and frosted). The cut sample was cleaned by subsequent sonications in acetone then isopropyl alcohol (IPA), and attached to a SHeM sample slide with an adhesive carbon tab. For the purposes of HAS measurements, separate ca. 7 × 7 mm pieces of each surface treatment were cut with a diamond cutter from the same glass slide as used for the SHeM measurements. After sonication, each sample was mounted to a HAS sample holder via a metal clamp.

### Sample preparation: Au on Si layers
A 2" diameter, 0.5 mm thick Si wafer from Sigma-Aldrich (<100>, single side polished, N-type, part number: 647780) was cleaved into ca. 7 × 5 mm, ca. 7 × 12 mm, and ca. 12 × 12 mm substrates. After rinsing with acetone and IPA, the samples were placed onto cellulose acetate (samples in Fig. 3) or poly(methyl methacrylate) (samples in Fig. 4) shadow masks. Various thickness Au layers were evaporated with an Angstrom Engineering thermal evaporator using A&E Metals 24ct Fine Gold Granulate (GGR24Y) onto the individual masked Si wafer substrates. Corresponding thermal evaporation rates can be found in Table 1. The ca. 7 × 5 mm samples were used for SHeM imaging; adhered to a SHeM sample slide in batches of 4. The ca. 12 × 12 mm samples were affixed to a SHeM sample slide individually. The ca. 7 × 12 mm substrates were used for AFM, SEM, and HAS measurements, with each different layer thickness sample adhered to the relevant instrument sample holder. For the patterned Au films shown in Fig. 1, the same evaporation process and Si wafer substrate was used to produce 3 different Au thickness samples. Each substrate was masked by a TEM grid (PELCO, 8GC-90), resulting in unaffected silicon (bright regions) and the gold film (darker regions). All samples were attached to their specific sample holders via adhesive carbon tabs.

### SHeM image acquisition: instrument
Further details concerning the experimental apparatus used to collect the neutral helium micrographs in this manuscript can be found in the literature[6,37]. The helium source was operated at 200 bar, 297 K stagnation temperature and with a 10 μm nozzle, with the resultant free-jet expansion progressively apertured by a Beam Dynamics skimmer (Model 2, 100 μm diameter) and a pinhole (5 μm diameter). During imaging, the sample chamber pressure was typically of the order $1 \times 10^{-8}$ mbar. Image acquisition is specified in terms of a dwell time, consisting of a wait (time allocated for the system to reach equilibrium) and a read (actual measurement time of the signal) component, in line with standard stagnation detector operation for neutral atom beams[38]. For all subsequent descriptions of SHeM micrographs, the dwell times are specified as wait/read. Although current recording times for the presented SHeM images can span up to several days, there are two important factors to be considered to put these times into perspective: (a) at this early stage of SHeM imaging technology development, the limiting factors in imaging times are the current neutral helium detector sensitivities and response times. Work is ongoing on improving neutral helium detectors[39–41] as well as developing instruments utilising more than one detector[42,43] decreasing imaging times by several orders of magnitude. (b) when considering the SHeM imaging time required to investigate sub-nanoscale features over areas of several $mm^2$ it is important to note that other techniques (e.g., SEM) would require much longer imaging times (~weeks/months/years) to cover the same area at the resolution required to visualise sub-nanoscale features.

### SHeM image acquisition: Fig. 1
The micrographs of the masked gold layers were collected using a 600/3000 ms dwell. For micrograph 'a' this total dwell was achieved by summing two component images, each with 600/1500 ms dwell.

**Table 1 | Thermally evaporated Au layer thickness**

| Au layer thickness (Å) | Thermal evaporation rate (Å/s) |
| --- | --- |
| 1 ± 0.2 | 0.05 |
| 5 ± 1 | 0.1 |
| 9 ± 2 | 0.8 |
| 26 ± 6 | 2 |
| 40 ± 10 | 1 |
| 80 ± 20 | 2 |
| 235 ± 58 | 2 |
| 778 ± 190 | 3 |
| All evaporations in Fig. 4 | 0.1 |

### SHeM image acquisition: Fig. 2

SHeM data for the glass material system was collected in a single micrograph with 1000/1500 ms dwell. An image with a suitable background region for contrast calculations was conducted with identical dwell. The diamond material system used 5 component images, each with 1200/1800 ms dwell, which were then summed to produce the final micrograph. An image with a suitable background region was conducted with 1200/9000 ms dwell. Errors in the contrast are estimated using the standard deviation of the intensity in the regions used to calculate contrast and standard error propagation.

### SHeM image acquisition: Fig. 3

To avoid potential detector drift, multiple SHeM micrographs were recorded for each gold layer thickness and then combined to generate the final image. In particular, for each SHeM micrograph shown, 15 separate SHeM scans were collected using 650/2000 ms dwell per pixel and then summed. An image with a suitable background region for contrast calculations was also conducted after each image set using a 1000/30,000 ms dwell. Errors in the contrast are estimated using the standard deviation of the intensity in the regions used to calculate contrast and standard error propagation.

### SHeM image acquisition: Fig. 4

Micrograph 'a' used 1000/8500 ms dwell, while micrograph 'c' employed 500/1500 ms dwell.

### SHeM image acquisition: Supplementary Fig. 4

For the temperature studies in Supplementary Fig. 4, the microscope was run as detailed previously except that the beam stagnation volume and nozzle were heated to alter the produced beam energy (see Supplementary Fig. 4 for the specific temperatures used in each study). The micrographs for the glass and diamond studies used 1000/1500 ms dwell, with background regions collected after each image using identical dwell. The gold study was conducted using the (26 ± 6) Å Au on Si sample seen in Fig. 3. To avoid detector drift, for each stagnation temperature five separate SHeM scans were collected using 1200/2000 ms dwell per pixel and then summed. An image with a suitable background region for contrast calculations was also conducted after each image set using a 1200/10000 ms dwell.

### SHeM image acquisition: Supplementary Fig. 14

To avoid potential detector drift, multiple SHeM micrographs were recorded and then combined to generate the final image. For the micrograph shown as the inset in Supplementary Fig. 14, five separate SHeM scans were collected using 1000/1200 ms dwell per pixel. An image with a suitable background region for contrast calculations was also conducted after each component image (5 total) using a 1000/1200 ms dwell.

### AFM image acquisition

All AFM images were recorded on an Oxford Instrument Asylum Research Cypher atomic force microscope in Tapping mode (standard topography/AC Air top) with a Budget Sensor Tap300Al-G tip. A range of AFM images were recorded on the glass, diamond, and Au on Si samples. Note that the same samples were used for both AFM and HAS experiments. For the Au on Si samples, reference AFM images were recorded on the pure silicon surface for each of the different layer thicknesses to ensure consistency across the dataset. Detailed AFM analyses of the glass and diamond model surfaces (including 1- and 2D power spectral densities, height, and slope distributions) were calculated using the open-source microscopy suite Gwyddion (www.gwyddion.net) and are provided in Supplementary Discussion 2.

### SEM image acquisition

All SEM images were recorded on an eLine system from RAITH with the Everhart-Thornley detector (SE2) and a 10 keV probe beam.

### HAS measurements: instrument

All HAS measurements were performed using the molecular beam scattering apparatus located at the University of Bergen, henceforth referred to as MAGIE[44]. A sketch of the experimental setup used in MAGIE is shown in Fig. 5. A description of the base operating principles for HAS instruments as well as a schematic describing the TOF principle can be found in[13]. The neutral helium beam was generated by a free-jet expansion from a (10 ± 1) μm nozzle (Plano GmbH, A0300P) at a stagnation pressure of (80 ± 1) bar. Note, this stagnation pressure was lower than that for the SHeM (200 bar), since MAGIE is not equipped for long term measurements at such high pressures. The stagnation temperatures for the Au on Si, diamond, and glass measurements were 300 K, 300 K and 305 K, respectively. The elastic energy of the helium beam, $E_i$ was evaluated experimentally from a TOF measurement on the direct beam. It was found to be $E_i = 67.20 ± 0.25$ meV and $E_i = 68.25 ± 0.25$ meV for the 300 K and the 305 K beam respectively. All samples were kept at room temperature ($T_s = 294.5 ± 1$ K) during the measurements. To mimic the SHeM conditions as closely as possible, no sample cleaning procedures were performed. The base pressure in the sample chamber during operation was around $3 × 10^{-8}$ mbar. The helium beam spot size on the sample was ca. 4 mm for all experiments. The intensity was recorded using a home-built ionisation detector[44] with an opening of 4.6 × 6.6 mm. Detector sensitivity changes between the measurements were accounted for by calibration to the same background signal for each measurement. No data were excluded from the analysis. All HAS measurements were performed on one set of samples.

### HAS measurements: rocking curves

For rocking curve measurements, the source-to-detector angle ($\alpha_D$) was fixed at 90°, and the chopper disc was removed from the beam line. Rocking curves were then recorded by varying the incoming beam angle $\theta_i$ (for MAGIE, $\theta_i = \alpha_m$, where $\alpha_m$ is the manipulator angle) relative to the sample surface normal and measuring the total reflected intensity. Reproducibility of the rocking curves was ensured by recording measurements at the same parameter set repeatedly at different times for the glass and the Au on Si sample sets. For the Au on Si samples each layer thickness had its own Si reference surface on the same sample providing for an additional reproducibility check on the Si results. For the diamond sample set just one set of measurements was performed. The total number of recorded rocking curves for the various samples were as follows: 5, 6, and 3 for smooth, frosted, and sandblasted glass (respectively); 1 for each of the micro and intermediate-diamond samples; 2, 3, and 2 for Au on Si samples of 9 Å, 26 Å, and 235 Å thicknesses (respectively).

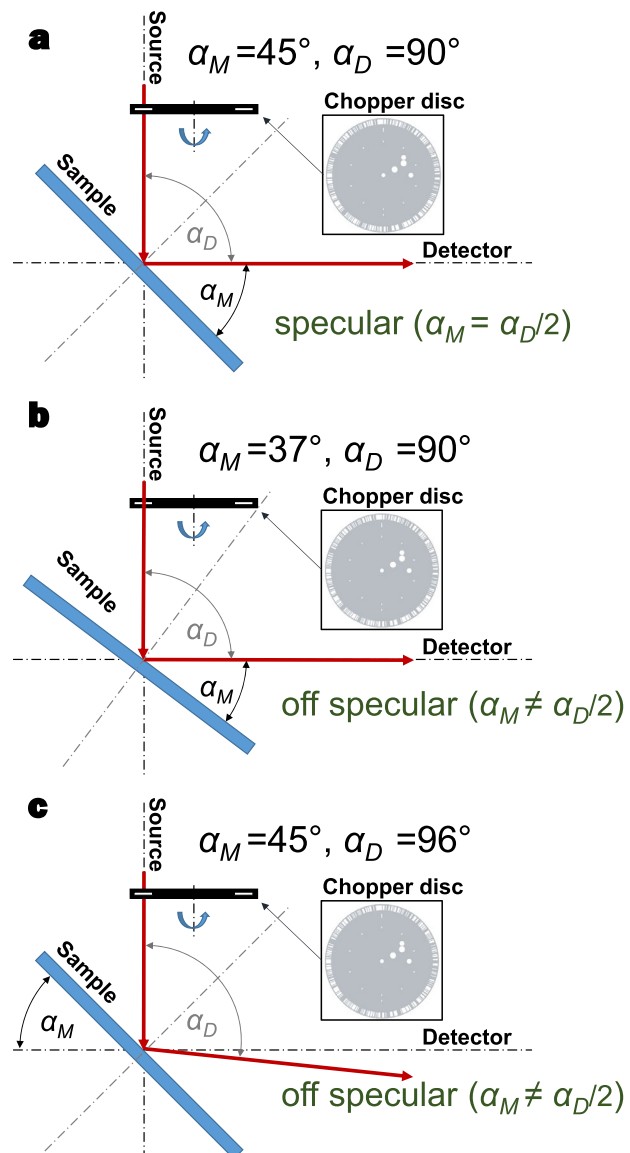

**Fig. 5 | MAGIE experimental setup.** Schematic of the experimental arrangement used in MAGIE in terms of the source-to-detector angle ($\alpha_D$) and the manipulator angle ($\alpha_M$). **a** Arrangement used for specular TOF measurements, where the incoming beam angle equals that of the outgoing beam with respect to the surface normal. Off-specular TOF measurements can be made either by rotating the sample relative to the incoming beam through changing $\alpha_M$ (**b**), or by rotating the detector relative to the incoming beam by varying $\alpha_D$ (**c**).

## HAS measurements: TOF

Energy-resolved spectra were obtained using TOF; recording the time-resolved intensity of reflected beam packages generated by mechanical chopping using a pseudorandom chopper (chopper rotation frequency 450 Hz)[45]. The chopper disc to sample distance was $(1003 \pm 3)$ mm and the sample to detector distance was $(902 \pm 3)$ mm. Knowing the initial energy of the atoms, the path length they travel from the chopper to the sample, and the path length from the sample to the detector, the arrival time can be converted into a velocity and thus an energy. Specular measurements ($\alpha_M = \alpha_D/2$) were recorded at $\alpha_D = 90°$ and $\alpha_M = 45°$, where again for off-specular measurement conditions ($\alpha_M \neq \alpha_D/2$) either $\alpha_D$ was slightly increased or decreased (with $\alpha_M = 45°$) or $\alpha_D$ was kept at 90° and $\alpha_M$ was decreased or increased. TOF measurement spectra where then transformed into energy-resolved spectra. The final energy spectra

are presented as an energy change of the reflected atoms relative to the initial energy of the atoms in the beam. All TOF measurements were recorded over 511 channels with a channel width of $4.35\,\mu s$, a delay time of $d_t = 12\,\mu s$ and a measurement time of $t_m = 3000$ s. Based on these settings, each TOF channel (time channel) is addressed $1.35 \times 10^6$ times per measurement. To ensure reproducibility, several TOF curves were recorded for the glass and the Au on Si sample sets at the same detector and manipulator positions at different times. For the Au on Si samples each layer thickness had its own Si reference surface on the same sample providing for an additional reproducibility check on the Si results. For the diamond sample set just one set of measurements was performed. The total amount of recorded TOF files for the various sample sets were as follows: 10, 12 and 10 for smooth, frosted and sandblasted glass (respectively); 4 for each of the micro and intermediate-diamond samples; 8, 15 and 6 for Au on Si samples of 9 Å, 26 Å, and 235 Å thicknesses (respectively). After accounting for detector sensitivity changes, the redundancy measurements at the same detector and manipulator position concurred well with each other. Note that for clarity, representative TOF measurements are presented in Supplementary Discussion 1; all other experimental datasets collected concurred with the presented data.

## Modelling

The facet-scattering model (MATLAB) estimates experimental SHeM micrograph intensities using the height maps collected or estimated from AFM or SEM (respectively). The following section describes the logical process by which the model simulates SHeM intensities.

The SHeM scattering geometry is used to generate a lookup table that encodes the probability for a ray to scatter specularly from a single, isolated facet into the detector (as a function of local tilt angle) via ray tracing. Next, an appropriate height map is imported and scaled in all three Cartesian axes. Using nearest neighbours interpolation, the local gradient—and thus tilt angle at each facet—is obtained. To model the effects of shadowing[6], the imported height map is rastered line-by-line underneath a fixed 'beam', with the facets that are struck by the incoming rays stored in a 'hit matrix' (thus taking into account the effective area of the individual facets).

The hit matrix, local facet angle, and the lookup table determine which particular facets lead to outgoing rays striking the detector. The outgoing, specularly scattered ray is tracked through the height map (with periodic boundary conditions) to see if a tall feature intersects it prior to entering the detector (masking[6]). If no such features block the path of an outgoing ray, only the local facet angle will determine whether the ray enters the detector and is marked as a 'hit'; however, if masking occurs the hit is nullified. The final intensity is then calculated by counting the relative fraction of incoming rays that strike the detector aperture after interacting with the sample. For each imported height map, the simulation is run for the map rotated at angles of 0, 90, 180, and 270 degrees. The mean and standard deviation of the relative intensities for all rotations are recorded, with standard error propagation then utilised to compute the datapoints and error bars given in Figs. 2 and 3. Repeating the analysis for different images of the same surface reveals that the uncertainties associated with the height fields represents approximately 50 to 90% of the total error across the Au on Si, diamond and glass model systems.

## Data availability

The datasets generated during and/or analysed during the current study are available from the corresponding author on request.

## Code availability

The code generated during the current study is available from the corresponding author on request.

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

## Acknowledgements

We thank Mr. Thomas Myles and Mr. Chris Hatchwell (University of Newcastle, Australia) for their assistance in the collection of several SHeM datasets, Dr. Nicolas Nicolaidis and Dr. Matthew Bergin (Australian National Fabrication Facility, Materials Node) for useful discussions, Dr. Benjamin Vaughan for collecting several AFM micrographs (Australian National Fabrication Facility, Materials Node), Dr. Justas Zalieckas (University of Bergen, Norway) and Dr. Vahid R. Adineh (Australian National Fabrication Facility, Victorian Node) for the production of the

investigated diamond samples. This research was supported under the Australian Research Council's Discovery Projects (Project No. DP170103979, P.D.) funding scheme. S.D.E. has received funding from the Research Council of Norway through a FRIPRO Mobility Grant (contract no. 250018/F20, S.D.E.). The FRIPRO Mobility grant scheme (FRICON) is co-funded by the European Union's Seventh Framework Programme for research, technological development and demonstration under Marie Curie grant agreement (no. 608695). This work was performed in part at both the Melbourne Centre for Nanofabrication (MCN) in the Victorian Node of the Australian National Fabrication Facility (ANFF), as well as the Materials node of the Australian National Fabrication Facility, which is a company established under the National Collaborative Research Infrastructure Strategy to provide nano- and micro-fabrication facilities for Australia's researchers.

## Author contributions

All sample preparation was performed by S.D.E. and M.G.B.; calibration of the thermal evaporator was carried out by S.D.E. and M.G.B.; A.F., S.D.E. and M.G.B. collected the SHeM data; S.D.E. collected the AFM and SEM data; experimental contrast analysis was conducted by A.F., M.G.B. and S.D.E.; facet-scattering model was conceptualised by A.F., S.D.E. and M.G.B.; practical implementation of the model was carried out by M.G.B.; S.D.E collected the experimental HAS data and the data analysis presented in Supplementary Discussion 1 was done by S.D.E., J.R.M. and B.H.; M.G.B. conducted the analysis highlighted in Supplementary Discussion 2; A.F. and M.G.B. conducted the analysis in Supplementary Discussion 3; A.F., S.D.E., M.G.B. and P.C.D. wrote the manuscript with input from all co-authors; A.F., S.D.E. and M.G.B. prepared the Figures; and J.R.M. and B.H. provided critical review and commentary in the revision of the manuscript.

## Competing interests

The authors declare no competing interests.
