## [Peer Review File · Nature Communications]

Reviewer comments, first round -

Reviewer #1 (Remarks to the Author):

See the attached pdf file.

Gregor Hlawacek

Reviewer #2 (Remarks to the Author):

Paper presents comparison of Scanning Helium Microscopy images on rough surfaces with a novel model of the contrast formation based on use of diffuse elastic scattering as a physical model and on processing independently measured height fields to simulate the microscope response. The methodology is sound, original, paper is clearly written and there is a large supplementary information provided to support the choice of the interaction model. Paper can have significant impact on the field. There are however two aspects that could be still improved:

- when dealing the measured data to simulate the Michelson contrast, e.g. in graph 3, the error bars are obtained from multiple model runs on rotated height fields. This would mean that the uncertainties of the height fields (as coming from the microscopes) are not propagated further. Taking into account that the feature size is very small on the studied surfaces, such data can be influenced by the data acquisition process, like tip convolution in AFM, which is known to have potential of largely impacting the surface normal statistics. It would be good to guess how much the experimental data uncertainty would affect the error bars in Fig. 3.

- the title and abstract gives a feeling like there is some super-resolution demonstrated in the paper. This might be misleading. The paper presents explanation how sub-resolution features form the contrast in SHeM, but wording "sub-resolution contrast" might be not ideal as it can lead both to correct and incorrect interpretation.

Reviewer #3 (Remarks to the Author):

The paper describes the use of scanning helium microscopy to evaluate nanoscale topography over large areas. The use of helium ion microscopy on a broad range of applications has been most welcome by the scientific community. The authors efforts in demonstrating another application of helium ion microscopy will add to the usefulness and importance of this technique and is highly commendable. However, while the results show sensitivity to nanoscale topography, this work does not show that quantitative surface topography can be extracted using the methods presented. Although this research is important, I recommend that the manuscript be rejected based on the information provide below.

General Comments:

The authors selected an extremely complicated surface to use in validating their technique. A sample with simple periodic structures within the range of interest is unambiguous, and much better for the type of technique validation presented in the manuscript.

In addition, it is not clear that the roughness values are comparing the same thing. For example, based on the scale bars shown in fig. 2, the area covered by SHeM micrographs are orders of magnitude larger than that of the AFM. Roughness values do not scale in this way. A necessary condition for this type of comparison is to determine and state the spatial frequency range of interest. This information is not included anywhere in the manuscript.

Other comments:

- For completeness, state the depth of surface region that the SHeM is sensitive to.
- Figures 2 and 3, and other places in the text: For consistency, please report the thickness values in nm or μm .
- "The averaged linescans (Fig. 1 d - e) highlight that the measured contrast does not change monotonically with thickness, but rather is greatest for the 80 Å film."

Comment on why you think the sample in Fig.1e would have the highest intensity. Although the statement about lack of a monotonic change is correct, it does not adequately capture the information here. The main information from these images is that there is no correlation between the height of these samples and the measured intensity.

Should it be "Fig. 1 d-f" in the above sentence? See Fig. 1 caption for additional correction if needed.

- "The facet scattering model (detailed in Methods) utilises height maps, such as those generated by an atomic force microscope (AFM) to calculate the local tilt at each pixel via nearest neighbour interpolation."

The Local tilt at each pixel will be highly dependent on tilt angle of the AFM and other probe parameters and will vary from probe to probe. The uncertainty of the probe parameters must be well within the tolerance needed to determine "detector acceptance" for this to work.

- Fig 2.: What is the spatial frequency range for the roughness values? Without the same spatial frequency range, the roughness comparison is meaningless.

- Define QCM on first use.

- Fig. 2 caption: "... Areal RMS roughness (S_q) was derived from the AFM micrographs of each surface (orange bar chart), with the axis scaled to best match the experimental SHeM contrast."

What does "... with the axis scaled to best match the experimental SHeM contrast" mean in the sentence above. Please comment.

- "Second, a variety of 2D image filters were applied to the AFM maps prior to running the simulations. Whilst this image processing reduced computational runtime, it also destroyed the predictive power of the model in every case."

The above statement is too general. What types of filters were applied to the AFM image? How were the images attenuated by these filters? What types of results do you get without the filters?

- Fig. 3, center figure: Comment on what the fit (red dotted line) indicates?

- "All AFM images have been restricted to the same amplitude range (0-3.06nm) for direct comparison."

By restricting the AFM scan in this way, if there are peaks or tall structures (deviations) on the surface, they will not be captured by the AFM, thereby biasing the results.

- Fig. 4: How do corresponding AFM images compare? While such a comparison with these specific features may not be possible because of differences in instrument scan sizes, using samples with well-defined features will help readers assess the robustness of the method.

Reply to the Reviewers' comments:

We thank the reviewers for their insightful comments. The following is a detailed description of the changes to the manuscript based on these comments. All issues raised by the reviewers have either been addressed directly in the text or discussed at length in this document. The original reviewer's comments are presented in bold, followed by our response in plain text. Changes and additions to the text of the manuscript are presented in italics.

Reviewer 1:

Summary: The paper describes the relation between the sample surface roughness and the signal intensity in unique Helium atom microscope. Several samples with different surface roughness are investigated and a model is developed that explains the observed change in detector signal. The model is based on the slope distribution on the sample surface and uses ray tracing to predict the amount of Helium atoms reflected into the detector.

It appears that there may be some misunderstanding by the reviewer regarding the discussion in the paper. The paper describes the unique sensitivity of the reflected He signal intensity to the atomic-scale surface topography. Indeed, we discuss in detail that the reflected He signal intensity is not simply correlated with the surface roughness.

Abstract: While all of this is correct I am not convinced about the reversibility of the process and even less convinced on the proposed use case of film thickness characterization in a general way.

We thank Reviewer 1 for their constructive feedback on the manuscript. We address the detailed concerns that the reviewer has regarding the concept of SHeM contrast inversion introduced in this paper, and its application to Ångström-scale film thickness characterization, in our detailed responses to the reviewer's comments.

1. The first one requires a unique relationship between the surface roughness and the observed signal intensity. The slope distribution alone is not sufficient to describe a regular or random rough surface. The authors put some effort into the characterization of the rough surface but I suggest to include a detailed characterization of the surface roughness including the Hurst parameter and the lateral correlation length.

Again, there may be some misunderstanding by the referee here, namely that the paper is specifically focussed on the relationship between surface topography (rather than simple roughness) and SHeM contrast.

However, we agree with the referee that a simple surface roughness metric is insufficient to fully describe surface topography. To address this comment, an extensive investigation regarding the characterisation of the rough surfaces was performed. In addition to the 2D RMS roughness, the correlation length and Hurst parameter are now presented for each surface based on the collected AFM maps with height and

slope distributions included in Extended Data. In summary, while the RMS roughness, correlation length and Hurst values calculated from the AFM map data correlate with the SHeM image contrast for the model glass dataset, they are wholly insufficient in explaining the model diamond dataset.

As such, the dependence of the image formation process upon surface topography cannot be described by a simple topography metric, whether RMS roughness, Hurst parameter or correlation length.

In response to these comments, we have made the following changes to the manuscript:

- a) Two extra plots have been added to Figure 2, presenting the Hurst parameter and lateral correlation length.
- b) The following text (describing the Hurst parameter and clarifying its use in validating the topographic length scales encompassed by the model surfaces) has been added

Various surface treatments (on glass samples) and growth conditions (for diamond samples) were used to generate model surface topographies with feature sizes spanning the nano to micro length scales (see Methods section for full details). The Hurst parameter (H) describes the nature of the 'roughness' of a data series – the smaller the value of H the less long-range order and vice versa. When H is exactly 0.5, the data can be described by geometric Brownian motion; $H < 0.5$ corresponds to little surface waviness, ie a jagged surface profile; and $H > 0.5$ indicates longer range order, such as faceting [15]. AFM height maps (Fig. 2a and extended data) confirmed that the surface features spanned a broad range of Hurst parameters consistent with the desired surface topographies.

To further clarify that simple topography metrics (whether RMS roughness, Hurst parameter or correlation length) for the diamond model system are not correlated with the He signal intensity (and thus a more sophisticated facet-scattering model is required) the text describing Figure 2 now reads:

As the AFM maps are of the order the helium spot size on the sample, a possible explanation for the observed differences in SHeM images is that the sub-resolution contrast arises simply from localised variations in 2D RMS roughness (S_q). Fig. 2c and 2d compare the simulated and experimental SHeM contrast to S_q as a function of length scale for both glass and diamond systems. The relative SHeM contrast was confirmed using complementary HAS scattering measurements (see Methods section and Extended Data). From Fig. 2, we see that while the RMS roughness values calculated from the AFM map data correlate with the SHeM image contrast for the glass dataset, they are wholly insufficient in explaining the diamond dataset. Higher order moments (such as skewness and kurtosis) were also found to be incapable of explaining the observed trends in SHeM contrast. As such, the dependence of the image formation process upon surface topography cannot be explained by simple metrics of surface topography [16]. For example, the S_q values shown in Fig. 2d indicate that the micro-diamond surface should yield the most contrast; yet the intermediate-diamond surface appeared darkest in SHeM micrographs. More sophisticated analysis methodologies such as the Hurst parameter and the correlation length (Fig. 2e and 2f) were also unable to explain the SHeM contrast for the diamond system (see Appendix 2 for the specific calculation).

By comparison, there is good agreement between the simulated and experimental SHeM contrast for both materials systems, demonstrating that image contrast is a direct result of the interplay between the helium beam and the specific sample topography. As it is both the distribution of surface features and the scattering geometry working in concert that yield the contrast apparent in SHeM micrographs, only the facet scattering model can replicate the intricacies of the image formation process. Moreover, the observed trends in the SHeM contrast data for the single component surfaces are more complex than simple roughness metrics and statistical moments indicate.

c) In addition, we added our extensive analysis on the Hurst parameter, aspect ratio and correlation length to the Extended Data section of the manuscript.

2. The latter proposed application to the determination of film thickness is misleading as it is based on a well-known change in roughness.

We strongly disagree with the reviewer that the proposed application of SHeM contrast to the determination of film thickness is somehow “misleading”. Instead, the paper shows that for materials systems whose topography as a function of film thickness is well-characterised (such as Au/SiO₂), the He contrast can be fully explained by an atomic facet scattering model thereby enabling the reflected He signal to be calibrated as a function of film thickness.

3. In addition even in this very special use case for the growth of gold thin films on SiO₂ substrates the relation between film thickness and Michelson contrast is not unique (see figure middle plot). A general application to the “evaluation of nano-coatings and thin films across large areas” is unlikely.

While we completely agree with the reviewer that for the Au/SiO₂ system the relation between film thickness and Michelson contrast is not unique, we disagree with the assertion that a general application to the evaluation of nano-coatings and thin films across large areas is unlikely. There are many characterisation techniques (e.g. ellipsometry [1,2], interferometry [3], computer tomography [4,5]) that deliver non-unique solutions. Indeed, almost any reconstruction technique faces this issue. The solution is to ensure that the technique is working within the constraints that allow a unique solution to be provided. For the Au/SiO₂ system, constraining the Au film thickness to $\lesssim 20 \text{ \AA}$ ensures a unique solution.

1. Hilfiker, A. N. et al. Survey of methods to characterize thin absorbing films with Spectroscopic Ellipsometry. *Thin Solid Films* **561 (22)**, 7979-7989 (2008).
2. Kats, M., Blanchard, R., Genevet, P. et al. Nanometre optical coatings based on strong interference effects in highly absorbing media. *Nature Mater* **12**, 20–24 (2013).
3. Park, Y., Depeursinge, C. & Popescu, G. Quantitative phase imaging in biomedicine. *Nature Photon* **12**, 578–589 (2018).
4. Levine, Z. H Nonuniqueness in dual-energy CT. *Med. Phys.* **44 (9)**, 202-206 (2017).
5. Wang, G., Ye, J.C. & De Man, B. Deep learning for tomographic image reconstruction. *Nat Mach Intell* **2**, 737–748 (2020).

To clarify the constraints used to ensure a unique solution, the text describing the inversion of the derived image contrast function now reads:

The extreme sensitivity of SHeM contrast to nanoscale topographic features, in combination with its non-damaging nature, enables sub-resolution defect detection across large areas. In particular, for a well-characterised material system, inversion of the derived image contrast function (Fig. 3) can be used to examine layer quality across the breadth of the sample. Although the relation between film thickness and Michelson contrast is non-unique, consistent with most reconstruction-based techniques used to evaluate nano-coatings and thin films (e.g. ellipsometry [32,33], interferometry [34] and computer tomography [35,36]) working within appropriate constraints (film thickness $\lesssim 20 \text{ \AA}$) allows a unique solution to be provided.

4. Considering the not very well explained time efforts (several hours at very low pixel resolution) of the method the final statements of the authors seem even less realistic.

While we acknowledge the current long SHeM imaging times, we disagree with the reviewer's assessment that the forward-looking final statements in the paper are unrealistic.

First, at this early stage in the development of neutral atom microscopy, the limiting factors in imaging times are the current neutral helium detector sensitivities and response times. As such, to reduce imaging times, an increase in available helium signal is required. Work is ongoing on improving neutral helium detector sensitivities [6 – 8], optimising neutral helium optics [9 – 13] as well as on optimising source designs for higher beam intensities [14]. In addition, several instruments are being developed utilising more than one detector, which will ultimately decrease the required imaging time [15,16].

Second, when considering the imaging time required to investigate sub-nanoscale features over areas of several mm², as presented in the SHeM images, it is important to compare with other state-of-the-art techniques like charged particle probe microscopes. For example: imaging the same surface area as Figure 4a (ca. 81 mm²), recorded with a magnification factor of 100,000 to ensure thin layers like the 1.2 Å Au layer are visible (consistent with the SEM images of the thin Au layers shown in the manuscript) will, with stitching the SEM scan areas together to cover the whole 81 mm², take about 37 years to record. This is under the assumption that 1 image takes 10s, including the move to the next stitching area. Moreover, the stitched image itself would be very time intensive to check for nanoscale changes, and probably would have to be constantly monitored. As such, the current SHeM imaging times are not particularly restrictive considering the current state-of-the-art.

6. Alderwick, A. R. et al. Simulation and analysis of solenoidal ion sources. *Rev. Sci. Instrum.* **79**, 123301 (2008).
7. Martens, J. et al. Development of a permanent magnet alternative for a solenoidal ion source. *Nucl. Instruments Methods Phys. Res. Sect. B Beam Interact. with Mater. Atoms* **340**, 85–89 (2014)
8. Bergin, M. Instrumentation and contrast mechanisms in scanning helium microscopy, PhD Thesis, University of Cambridge (2019). <https://www.repository.cam.ac.uk/handle/1810/290645>
9. Eder, S. D., Reisinger, T., Greve, M. M., Bracco, G. & Holst, B. Focusing of a neutral helium beam below one micron. *New J. Phys.* **14**, 073014 (2012)
10. Anemone, G., Taleb, A. A., Eder, S. D., Holst, B. & Farías, D. Flexible thin metal crystals as focusing mirrors for neutral atomic beams. *Phys. Rev. B* **95**, 205428 (2017).
11. Bergin, M., Ward, D. J., Ellis, J., Jardine, A. P. A method for constrained optimisation of the design of a scanning helium microscope, *Ultramicroscopy* **207**, 112833 (2019)
12. Palau, A. S., Bracco, G., Holst, B. Theoretical model of the helium pinhole microscope, *Phys. Rev. A* **94**, 063624 (2016)
13. Palau, A. S., Bracco, G., Holst, B. Theoretical model of the helium zone plate microscope, *Phys. Rev. A* **95**, 013611 (2017)
14. Even, U. Pulsed Supersonic Beams from High Pressure Source: Simulation Results and Experimental Measurements. *Adv. Chem.* **2014**, 1–11 (2014).
15. Lambrick, S.M. et al. True-to-size surface mapping with neutral helium atoms. *Phys. Rev. A.*, **102**, 053315 (2021)

16. Lambrick, S. M. The formation of contrast in scanning helium microscopy, PhD Thesis, University of Cambridge (2021). <https://www.repository.cam.ac.uk/handle/1810/336033>

To further clarify the issues associated with the SHeM imaging times:

- a) The following summary has been added to the Methods section of the Manuscript:

Although current recording times for the presented SHeM images can span up to several days, there are two important factors to be considered to put these times into perspective: (a) at this early stage of SHeM imaging technology development, the limiting factors in imaging times are the current neutral helium detector sensitivities and response times. Work is ongoing on improving neutral helium detectors [39,40,41], as well as developing instruments utilising more than one detector [42,43], decreasing imaging times by several orders of magnitude. (b) when considering the SHeM imaging time required to investigate sub-nanoscale features over areas of several mm² it is important to note that other techniques (e.g. SEM) would require much longer imaging times (~weeks/months/years) to cover the same area at the resolution required to visualise sub-nanoscale features.

- 5. I think this paper describes in an excellent way fundamental processes and the application of a unique and very interesting technique and should therefore be published in a more appropriate journal dealing with the application and development of scientific instruments. Considering the above statements I can not support the publication in Nature communication.**

We thank the reviewer for their very positive comments regarding the paper. However, given that the paper reports the application of a new atom-beam microscopy to a broad technological application of interest to a wide audience we believe that the manuscript is far more appropriate for publication in Nature Communications rather than a journal dealing with the specialised development of scientific instruments.

- 6. page 1; column 2; center A few sentences summarizing the cited literature on what limits the resolution of the method would help the reader.**

To clarify this issue the following text and reference has been added to page 1.

Currently, resolution is primarily limited by detector efficiency, with state-of-the-art detectors enabling resolutions of the order of 40 nm [7].

- 7. page 2; figure 2 Please explain the origin of the error bars.**

The origin of the simulation error bars in Figure 2 is given in the modelling section of the Methods section. The origin of the experimental error bars in Figure 2 was omitted from the original manuscript and has now been added to the SHeM image acquisition section of the Methods section.

To clarify this issue the following text has been added to the Figure 2 caption:

Details of the origin of the error bars are given in the Methods section.

and the following text has been added to the Methods section:

Errors are estimated from the standard deviation of the contrast.

- 8. page 3; column 1; top** As stated above the slope distribution alone does not seem to be sufficient to uniquely describe a rough surface. I suggest to also include the evaluation of the Hurst parameter and the lateral correlation length. In addition non-random rough surfaces could be used for a better understanding of the dependence of the signal intensity on the various roughness parameters. That the surface normal alone is not resulting in a unique relationship between surface roughness and Michelson contrast (signal intensity) can clearly be seen in the plot presented in figure 3 which has a several roughness values related to a single Michelson contrast value.

These points have already been addressed in detail in previous comments.

- 9. AFM data presented:** The analysis is based on the evaluation of topographic AFM data. However, the data presented is amplitude data. While this might be a good choice considering that this data shows slope, it is difficult to quantify. Presenting the actual used data alongside with the integrated slope distribution and the amplitude images might help the reader. Slope distributions from AFM data can easily be obtained using program like gwyddion.

The Extended Data now contains plots of the AFM data analysis including height and slope distributions as requested.

- 10. page 4; column 1; last paragraph** The inversion of the image contrast function is not unique and can therefore not be used for predictions.

This point has already been addressed in detail previous comments.

- 11. Final statement:** I am not convinced the proposed applications in quality control in connection with high throughput are realistic.

This point has already been addressed in detail previous comments.

- 12. Methods:** The pixel resolution for the presented SHeM is required to estimate the time required to obtain the data. Taking one of the shorter dwell times mentioned and a really bad pixel resolution of 128×128 pixels I arrive at a recording time for single image of more than 13 hours.

This point has already been addressed in detail previous comments.

Reviewer 2:

Paper presents comparison of Scanning Helium Microscopy images on rough surfaces with a novel model of the contrast formation based on use of diffuse elastic scattering as a physical model and on processing independently measured height fields to simulate the microscope response. The methodology is sound, original, paper is clearly written and there is a large supplementary information provided to support the choice of the interaction model. Paper can have significant impact on the field.

We thank Reviewer 2 for their constructive feedback on the manuscript. Reviewer 2 finds the work novel and appropriate for publication in Nature Communications as a means to expose the technique to a wider scientific audience.

There are however two aspects that could be still improved:

- 1. When dealing the measured data to simulate the Michelson contrast, e.g. in graph 3, the error bars are obtained from multiple model runs on rotated height fields. This would mean that the uncertainties of the height fields (as coming from the microscopes) are not propagated further. Taking into account that the feature size is very small on the studied surfaces, such data can be influenced by the data acquisition process, like tip convolution in AFM, which is known to have potential of largely impacting the surface normal statistics. It would be good to guess how much the experimental data uncertainty would affect the error bars in Fig. 3.**

We thank Reviewer 2 for their comment and agree that: (a) an estimate of the influence of the uncertainties of the experimental height fields upon the error bars in the model analysis would be a useful addition to the manuscript and (b) the height field uncertainties are likely generated by factors such as tip convolution.

To address (a), the total error arising from different images of the same surface generated using multiple model runs on rotated height fields was calculated. This analysis reveals that the uncertainties associated with the height fields represents approximately 50% to 90% of the total error. The modelling section of the Methods now reads:

For each imported height map, the simulation is run for the map rotated at angles of 0, 90, 180, and 270 degrees. The mean and standard deviation of the relative intensities for each rotation is recorded (yielding the datapoints and error bars given in Figs. 2 and 3). Repeating the analysis for different images of the same surface reveals that the uncertainties associated with the height fields represents approximately 50 to 90% of the total error across the gold, diamond and glass model systems.

To address (b), the following addition was made to the manuscript:

Second, when a 2D averaging filter or matrix downsampling was applied to the AFM maps prior to running the simulations (in an attempt to reduce computational runtime), the predictive power of the model was destroyed in every case. This image processing is analogous to collecting AFM micrographs with a blunt tip.

- 2. The title and abstract gives a feeling like there is some super-resolution demonstrated in the paper. This might be misleading. The paper presents explanation how sub-resolution features form the contrast in SHeM, but wording "sub-resolution contrast" might be not ideal as it can lead both to correct and incorrect interpretation. The paper appears to be technically sound and the methods are clearly described, with a comprehensive worked example included as an appendix which would greatly assist other researchers wishing to use the method. The claims made are fully supported by the experimental data and the figures in the main report and the appendix are well presented and aid in the understanding of the technique.**

The term 'sub-resolution' contrast in neutral atom microscopy has been previously established in the literature [1]. Moreover, the terms "supra-resolution" and "sub-resolution" as they pertain to the SHeM are clearly defined in the paper. This manuscript describes how the unique properties of SHeM imaging enable contrast changes for sub-resolution (below the instrumental lateral resolution) features to be observed.

1. Fahy, A. et al. Image formation in the scanning helium microscope. *Ultramicroscopy* **192**, 7–13 (2018).

Reviewer 3:

The paper describes the use of scanning helium microscopy to evaluate nanoscale topography over large areas. The use of helium ion microscopy on a broad range of applications has been most welcome by the scientific community. The authors efforts in demonstrating another application of helium ion microscopy will add to the usefulness and importance of this technique and is highly commendable. However, while the results show sensitivity to nanoscale topography, this work does not show that quantitative surface topography can be extracted using the methods presented. Although this research is important, I recommend that the manuscript be rejected based on the information provide below.

We thank Reviewer 3 for the time that they have taken to review the paper.

However, it appears that there is a significant misunderstanding by the reviewer regarding the focus of the paper with implications for the relevance of the review.

In particular, the reviewer appears to have confused neutral helium atom microscopy with the far more common technique of helium ion microscopy.

General Comments:

- 1. The authors selected an extremely complicated surface to use in validating their technique. A sample with simple periodic structures within the range of interest is unambiguous, and much better for the type of technique validation presented in the manuscript.**

The model glass and diamond surfaces provided the simple surface topographies (with feature sizes spanning the nano to micro length scales) needed to validate the facet scattering model. AFM height maps confirmed that the surface features spanned the required broad range of Hurst parameters ($H > 0.5$, $H = 0.5$, and $H < 0.5$). More specifically, the model surfaces comprised:

Sandblasted glass $H = 0.68$ (very large grains), Frosted glass $H = 0.61$ (smaller grains), Smooth glass $H = 0.21$ (very ragged changes between positions).

Micro-diamond $H = 0.61$ (large facets, smoother changes between positions), Intermediate-diamond $H = 0.47$ (geometric Brownian motion), Nano-diamond $H = 0.19$ (very ragged changes between positions).

A simple periodic structure (as suggested by the reviewer) would not span the required broad range of Hurst parameters and thus would be incapable of validating the facet scattering model. In addition, to the best of our knowledge it is currently not possible to fabricate periodic surface structures with feature heights ranging from sub-nanometer scale up to hundreds of microns made of a single material of the same material structure.

To clarify to the reader our choice of model samples, we added the following section to the manuscript:

Various surface treatments (on glass samples) and growth conditions (for diamond samples) were used to generate model surface topographies with feature sizes spanning the nano to micro length scales (see Methods section for full details). The Hurst parameter H (see Appendix 2) describes the nature of the

'roughness' of a data series – the smaller the value of H the less long-range order and vice versa. When H is exactly 0.5, the data can be described by geometric Brownian motion; $H < 0.5$ corresponds to little surface waviness, ie. a jagged surface profile; and $H > 0.5$ indicates longer range order, such as faceting [15]. AFM height maps (Fig. 2a and extended data) confirmed that the surface features spanned a broad range of Hurst parameters consistent with the desired surface topographies.

(To simplify the discussion, the following comments from Referee 3 have been grouped together as they all speak to the characterisation of the AFM data concerning the model surfaces and the appropriateness of using such maps as the basis for contrast modelling conducted in the manuscript)

2. In addition, it is not clear that the roughness values are comparing the same thing. For example, based on the scale bars shown in fig. 2, the area covered by SHeM micrographs are orders of magnitude larger than that of the AFM. Roughness values do not scale in this way. A necessary condition for this type of comparison is to determine and state the spatial frequency range of interest. This information is not included anywhere in the manuscript.

The referee appears to have misunderstood the fundamental relationships between the AFM maps, the scattering model and the SHeM micrographs. The referee is indeed correct that the AFM maps are smaller than the size of the SHeM micrographs, with a whole AFM micrograph fitting within a single SHeM pixel. However, the AFM and SHeM maps are not simply measuring the surface roughness at different scales. As we describe in the paper, the AFM maps quantify the characteristic surface topography for the atom beam scattering model, which in turn is used to predict the SHeM contrast. As such, it is the aggregation of all the individual sub-resolution scattering events that cause the "brightness" of a single SHeM pixel.

To further address this point, the Extended Data now contains plots of the AFM data alongside power spectral density showing the spatial frequency range of interest as requested.

3. Fig 2.: What is the spatial frequency range for the roughness values? Without the same spatial frequency range, the roughness comparison is meaningless.

To address this point directly, the Extended Data now contains plots of the AFM data alongside power spectral density showing the spatial frequency range of interest as requested.

4. Fig. 4: How do corresponding AFM images compare? While such a comparison with these specific features may not be possible because of differences in instrument scan sizes, using samples with well-defined features will help readers assess the robustness of the method.

Figure 4 presents the reconstructed SHeM images of the gold contacts evaporated onto silicon substrate wafers. The lateral range of these images is ~ 10 mm x 10 mm, which is far beyond that of most AFM systems, where typically a maximum scan area of (20 – 100) mm x (20 – 100) mm is imaged. Thus, as foreshadowed by the reviewer, there are no corresponding AFM images of these samples because of the large differences in instrument scan sizes. Indeed, assuming that it were possible to collect an AFM 20 mm x 20 mm image in 10 s, then the corresponding AFM image for Figure 4 would require stitching of 250,000 images and would take 29 days (694 hrs) to collect. Moreover, the thickness of the evaporated gold contacts is 10 ± 2 Å, which (as we show in Figure 3 and discuss earlier in the

paper) is in a gold coverage regime that the AFM is incapable of imaging due to gold adatoms adhering to the AFM tip. Thus, again we emphasize that it is the unique sensitivity of the incident He atoms to the sub-nanoscale surface topography that enables the detailed reconstruction of the height maps (and thereby the identification of Ångström-scale surface defects in thin films across millimetre lateral length scales) shown in Figure 4.

To further emphasise this point, the following text has been added to the discussion around Figure 4:

By comparison, a corresponding AFM image would require stitching of 250,000 individual (20 mm x 20 mm) images and (assuming 10 s per image) stable imaging for 29 days (694 hrs). Moreover, as discussed earlier, AFM imaging in this gold coverage regime (10 ± 2 Å) is unreliable due to gold adatoms adhering to the AFM tip.

Other comments:

5. For completeness, state the depth of surface region that the SHeM is sensitive to.

Unfortunately we believe this comment can be attributed to the reviewer's misunderstanding of the technique in question. The manuscript currently details the interaction between the neutral helium atom and the surface under investigation in the Abstract:

Since the helium atom scatters exclusively from the outermost electronic corrugation of the sample, the technique is completely surface sensitive.

As well as in the introductory paragraph:

As the probe cannot penetrate the bulk at all, the micrograph generated is exclusively of the surface under investigation.

6. Figures 2 and 3, and other places in the text: For consistency, please report the thickness values in nm or μm .

Figure 2 makes no reference to layer thicknesses as it is addressing SHeM contrast generation in the glass and diamond model systems.

For consistency, all of the gold layer thicknesses (as derived from the quartz crystal monitor) shown in Figures 3 and 4 are given in Ångströms. As such, we suggest respectfully that there is no reason to change the units to nm.

7. "The averaged linescans (Fig. 1 d - e) highlight that the measured contrast does not change monotonically with thickness, but rather is greatest for the 80 Å film." Comment on why you think the sample in Fig.1e would have the highest intensity. Although the statement about lack of a monotonic change is correct, it does not adequately capture the information here. The main information from these images is that there is no correlation between the height of these samples and the measured intensity.

Again, this comment highlights that there may be some misunderstanding by the reviewer regarding the conclusion from Figure 1. The reviewer is indeed correct that Figure 1 reveals that there is no simple relationship between the measured He atom contrast and the height of the thin gold film samples. Subsequently, the work discussed in Figures 2 and 3 in the manuscript shows that the contrast observed Figure 1 originates from sub-resolution topographic features.

To further clarify this point, the paragraph immediately following Figure 1 has been rewritten as follows:

In order to establish whether the non-monotonic contrast variations with gold thickness (observed in the SHeM micrographs shown in Figure 1) arise from sub-resolution topographic features, a bespoke helium atom – facet scattering MATLAB simulation was developed.

8. Should it be “Fig. 1 d-f” in the above sentence? See Fig. 1 caption for additional correction if needed.

References to “Fig. 1 d-e” have been updated to “Fig. 1 d-f” in the main text as well as the Figure 1 caption.

Define QCM on first use.

‘QCM’ has been defined as Quartz Crystal Microbalance in the Figure 3 caption and replaced with the full definition on Page 4 of the manuscript main text.

9. Fig. 2 caption: “... Areal RMS roughness (Sq) was derived from the AFM micrographs of each surface (orange bar chart), with the axis scaled to best match the experimental SHeM contrast.”. What does “... with the axis scaled to best match the experimental SHeM contrast” mean in the sentence above. Please comment.

The vertical axes in the plots are just scaled to allow a comparison between the trends in the roughness and the experimental/simulation SHeM contrast. The phrase has been removed as it is unnecessary.

10. Second, a variety of 2D image filters were applied to the AFM maps prior to running the simulations. Whilst this image processing reduced computational runtime, it also destroyed the predictive power of the model in every case.”. The above statement is too general. What types of filters were applied to the AFM image? How were the images attenuated by these filters? What types of results do you get without the filters?

Both a 2D averaging filter and matrix downsampling was applied to the AFM maps prior to running the simulations in an attempt to reduce computational runtime. However, neither the filter nor the downsampling were used since both degraded the simulations. Thus the results shown are without any filtering of the AFM images. In order to clarify this point the discussion has been amended to read:

Second, when a 2D averaging filter or matrix downsampling was applied to the AFM maps prior to running the simulations (in an attempt to reduce computational runtime), the predictive power of the model was destroyed in every case. This imaging processing is analogous to collecting AFM micrographs with a blunt tip.

11. Fig. 3, center figure: Comment on what the fit (red dotted line) indicates?

The red dotted line is included as a guide to the eye. To clarify this point the figure caption text now reads:

Centre: Plot of the experimental Michelson contrast between gold and silicon derived from the SHeM micrographs of the sample series as a function of QCM layer thickness. The predicted Michelson contrast between the gold and silicon from the facet scattering model is also shown, based on height maps derived from the SEM and AFM images as well as a line to guide the eye (red dotted line).

12. All AFM images have been restricted to the same amplitude range (0-3.06nm) for direct comparison.”. By restricting the AFM scan in this way, if there are peaks or tall structures (deviations) on the surface, they will not be captured by the AFM, thereby biasing the results.

We agree with the reviewer that the language concerning the amplitude range was ambiguous. The colourbar for each AFM image was not allowed to simply range between the minimum and maximum amplitudes for that particular scan; rather, the minimum and maximum amplitudes for the full data set were derived and the colourbar fixed to this range to allow for direct comparisons with no loss or biasing of the results. The same approach was taken for all comparative sets of AFM and SHeM micrographs to ensure no biases were introduced. To better communicate the methodology, the Figure 3 caption has been updated to read as follows:

All AFM images have been restricted to the same amplitude range (derived from the maximum amplitude found across the total dataset, namely 0 – 3.06 nm) for direct comparison.

Reply to the Editor's comments:

- 1. You will see that, while the reviewers find your work of interest, they raise substantive concerns that cast doubt on the advance your findings represent over earlier work and the strength of the novel conclusions that can be drawn at this stage.**

We thank the reviewers for their questions and comments. We have presented detailed and compelling arguments that address all the reviewers' concerns. In particular, this paper shows for the first time that neutral helium microscopy is capable of revealing Ångström-scale topography of thin-film coatings across lateral length scales of tens of millimetres; a capability that is either beyond (or impractical) for other microscopies. This finding represents a significant advance over our earlier work and which we demonstrate arises from the sensitivity of the incident helium atom beam to the atomic surface structure.

- 2. In particular, Rev.#1 and #3 expressed important technical concerns regarding the ability of the proposed method to quantitatively characterize the surface topography of thin films.**

We have addressed all the technical concerns raised by Rev. #1 and #3 in our responses given above.

- 3. Like the reviewers, we appreciate your demonstration of the sensitivity of scanning helium ion microscopy to the nanoscale topography, and we would be interested in a work that unequivocally shows Ångström-scale topography of thin-film coatings using this technique. On the whole, however, we agree with the comments of these reviewers that for this work to truly influence our analytical capabilities, it should demonstrate a unique relationship between the surface roughness and the signal intensity and a more general applicability to thin film coatings.**

We emphasise that the focus of this paper is the new technique of neutral helium microscopy and not the more established helium ion microscopy. The paper demonstrates that the contrast observed in neutral helium microscopy is directly related to the detailed atomic surface morphology for both model single material component systems and for more complex thin film coatings. Subsequently, the paper shows that by calibrating the contrast neutral helium microscopy does indeed unequivocally show ångström-scale topography of thin-film coatings.

Moreover, we reinforce the point (made in our earlier responses to Rev #1 and Rev #3) that demonstrating a unique relationship between surface topography [roughness] and signal intensity for all material systems and across the complete parameter space is generally not possible, nor is it required, for an imaging reconstruction technique to be extremely scientifically important. Instead, what is crucial is to ensure is that the technique is working within the constraints that allow a unique solution to be provided, which are typically based on reference measurements for specific material systems.

- 4. The reviewers suggest how you may fill this gap: adding characterization of further model surfaces, for instance with well-defined features or non-random roughness, additional characterization of the surface roughness, and careful revision of the AFM data analysis.**

We have addressed all the reviewers' comments and have provided detailed additional characterisation of the surface topography of the different material surfaces as well as a careful revision of the AFM data

Prof Paul Dastoor
Director, Centre for Organic Electronics
Physics
TELEPHONE: +61 2 4921 5426
FACSIMILE: +61 2 4921 6907
Email: Paul.Dastoor@newcastle.edu.au

analysis. This analysis confirms the conclusions of the original manuscript, namely that there is no single simple surface metric that correlates with the SHeM contrast whereas scattering simulations incorporating atom scale masking and shadowing of the incident helium beam are able to explain the form of the SHeM contrast data.

However, we reiterate the point (made in our earlier response to Rev #3) that adding characterization of further model surfaces such as a "simple periodic structure" would not span the required broad range of Hurst parameters and thus would be incapable of validating the facet scattering model. In addition, to the best of our knowledge it is currently not possible to fabricate periodic surface structures with ångström feature heights across millimetre lateral scales made of a single material of the same material structure.

We trust that the revisions that we have made satisfy the concerns of the reviewers and we look forward to hearing from you and the reviewers about the revised manuscript.

Yours Sincerely,

Paul Dastoor & Sabrina Eder (corresponding authors).

Reviewer comments, second round -

Reviewer #2 (Remarks to the Author):

All my questions raised in the previous report were addressed and paper was significantly improved (namely on basis of the other referee's comments), so I can recommend it now for publication.

Reviewer #3 (Remarks to the Author):

Many thanks for revising the paper, and for providing additional information. The quality of the paper and its readability is much better. Having said that, the need for an unambiguous comparison still holds. See clarifications below:

A couple of clarifications:

Question #1

It is correct that a simple periodic structure will not capture a board range of Hurst parameters, but that is the point. The goal is to do a basic validation of the technique using a single spatial wavelength and amplitude within a range both the AFM and SHeM can measure and show that the reconstructed height map is consistent with AFM data. No stitching is required. The scattering model would be simpler, and such a test would only show that the technique works for spatial the wavelength and amplitude and amplitude being measured. One measurement is enough, but you can show results for more than one sample if available.

Question #5

My wording for this question was ambiguous. This question refers to the height or depth of surface features, rather than penetration depth. The caption for figure 3 indicates that the technique can at least measure surfaces with feature heights of 3.06 nm. What is the upper limit of feature heights the technique can measure?

Reply to the Reviewers' comments:

We thank the reviewers for their insightful comments and work on reviewing our manuscript. The following is a detailed description of the changes to the manuscript based on these comments. All issues raised by the reviewers have either been addressed directly in the text or discussed at length in this document. The original reviewer's comments are presented in bold, followed by our response in plain text. Changes and additions to the text of the manuscript are presented in italics.

Reviewer 2:

All my questions raised in the previous report were addressed and paper was significantly improved (namely on basis of the other referee's comments), so I can recommend it now for publication.

We thank Reviewer 2 for this recommendation and the work on reviewing our manuscript.

Reviewer 3:

Question # 1: It is correct that a simple periodic structure will not capture a board range of Hurst parameters, but that is the point. The goal is to do a basic validation of the technique using a single spatial wavelength and amplitude within a range both the AFM and SHeM can measure and show that the reconstructed height map is consistent with AFM data. No stitching is required. The scattering model would be simpler, and such a test would only show that the technique works for spatial the wavelength and amplitude and amplitude being measured. One measurement is enough, but you can show results for more than one sample if available.

We have added a new appendix 3, called 'Basic Reconstruction', to the manuscript. This appendix presents a set of measurements performed on AFM height standards consisting of periodic line structures with varying height. These line structures were used to validate the height reconstruction technique on simple periodic structures. SHeM as well as AFM measurements were performed for this appendix.

In addition, we added the following sentence pointing towards the appendix to the manuscript. This sentence is leading into the section describing the reconstruction performed on Fig 4.:

A basic confirmation of such a reconstruction was performed using standard AFM calibration gratings (Appendix 3), before moving to the more industrially-relevant example of thermally evaporated polycrystalline gold contacts on a silicon substrate (Fig. 4).

Likewise, the Methods section as well as the Acknowledgements were updated corresponding to the additional experiments performed for appendix 3:

Fig. A3: To avoid potential detector drift, multiple SHeM micrographs were recorded and then combined to generate the final image. For the micrograph shown as the inset in Fig. A3, 5 separate SHeM scans

Prof Paul Dastoor
Director, Centre for Organic Electronics
Physics
TELEPHONE: +61 2 4921 5426
FACSIMILE: +61 2 4921 6907
Email: Paul.Dastoor@newcastle.edu.au

were collected using 1000/1200 msec dwell per pixel. An image with a suitable background region for contrast calculations was also conducted after each component image (5 total) using a 1000/1200 msec dwell.

We thank Mr. Thomas Myles and Mr Chris Hatchwell (University of Newcastle, Australia) for their assistance in the collection of several SHeM datasets, Dr. Nicolas Nicolaidis and Dr. Matthew Bergin (Australian National Fabrication Facility, Materials Node) for useful discussions ...

Question #5: My wording for this question was ambiguous. This question refers to the height or depth of surface features, rather than penetration depth. The caption for figure 3 indicates that the technique can at least measure surfaces with feature heights of 3.06 nm. What is the upper limit of feature heights the technique can measure?

As the referee points out, Figure 3 shows that when considering the sub-resolution contrast discussed in the paper, the technique can at least measure surfaces with feature heights of 3.06 nm. By comparison, then Figure 2 shows that feature heights of the order of several hundred nm can also be measured. Moreover, Figure 2 shows that the assumptions underpinning the generation of sub-resolution contrast break down when “supra-resolution features are the dominant contributor to contrast.” Thus the sub-resolution contrast arguments and model presented in this paper are most relevant for feature heights up to the lateral resolution of the SHeM instrument – in this case of the order of a few microns. Above this height, contrast due to supra-resolution features dominates and much larger feature heights (mm and above) can be readily measured from changes in mean plane scattering of the incident beam.

In order to clarify this point we have altered the summary text to read:

In summary, we have shown that the absolute surface sensitivity of neutral helium enables sub-resolution contrast in the SHeM to discern topographic features far below (and up to) the current lateral resolution of the instrument.

Reviewer comments, third round -

Reviewer #3 (Remarks to the Author):

Many thanks for revising the manuscript. The additional information and revisions adequately address my questions. I recommend publication.